# Stroke Risk in Young Women with Primary Dysmenorrhea: A Propensity-Score-Matched Retrospective Cohort Study

**DOI:** 10.3390/jpm13010114

**Published:** 2023-01-04

**Authors:** Chung-Hsin Yeh, Fung-Chang Sung, Chih-Hsin Muo, Pao-Sheng Yen, Chung Y. Hsu

**Affiliations:** 1Department of Nursing, College of Nursing and Health Sciences, Da-Yeh University, Changhua 515, Taiwan; 2Department of Neurology, Yuan Rung Hospital, Changhua 510, Taiwan; 3Management Office for Health Data, China Medical University Hospital, Taichung 404, Taiwan; 4Department of Health Services Administration, China Medical University College of Public Health, Taichung 406, Taiwan; 5Department of Food Nutrition and Health Biotechnology, Asia University, Taichung 413, Taiwan; 6Department of Neuroradiology, Kuang Tien General Hospital, Taichung 433, Taiwan; 7Graduate Institute of Biomedical Sciences, China Medical University, Taichung 404, Taiwan

**Keywords:** dysmenorrhea, propensity score, retrospective cohort study, stroke

## Abstract

Background: Studies on strokes associated with dysmenorrhea are limited. We conducted a propensity-score-matched retrospective cohort study to assess the risk of stroke in women with primary dysmenorrhea (PD). Methods: From the claims data of one million people in Taiwan’s insurance program, we identified 18,783 women aged 15–40 years, newly diagnosed with PD in 2000–2010, without a history of stroke. We randomly selected a comparison cohort without stroke history and dysmenorrhea, with the same sample size matched by age, index date, and propensity score. We began a follow-up with individuals one year after cohort entry to the end of 2013 to capture stroke events. Results: The two study cohorts were well-matched for age and comorbidities, with 54% of women aged 15–24. Stroke incidence was 1.5-fold higher in the PD cohort than in the comparison cohort (6.05 vs. 4.01 per 10,000 person-years, or 99 vs. 65 cases), with an adjusted hazard ratio (aHR) of 1.51 (95%CI 1.11–2.06) after adjustment for matched pairs. Nearly 70% of strokes were ischemic strokes, which occurred 1.6 times more frequently in the PD cohort than in the comparison cohort (4.40 vs. 2.71 per 10,000 person-years, or 72 vs. 44 cases), aHR = 1.61 (95% CI 1.11–2.33), after adjustment for matched pairs. The incidence of hemorrhagic stroke was also higher in the PD cohort than in the comparison cohort (1.65 vs. 1.29 per 10,000 person-years, or 27 versus 21 cases), but the difference was not significant. Conclusion: Women of reproductive age with PD are at increased risk for ischemic stroke.

## 1. Introduction

Women go through a menstrual cycle due to the monthly change in estrogen production. The menstruation period begins with menarche and ends after menopause. Dysmenorrhea is a painful condition for women during menstruation in which they experience an intense sensation of pain or even cramping in the lower abdomen. Adolescents and young women with severe and frequent cramps and pain from dysmenorrhea tend to be regularly absent from school and work [1,2]. Absence from daily work during the dysmenorrhea cycle may be responsible for the loss of 600 million work hours in the United States [3].

A previous review of the international literature of 178 studies found that the prevalence of dysmenorrhea varied widely by ethnic group, ranging from 16.8% to 81% [4]. The age of women is inversely related to the prevalence, which is higher in young women aged 17–24 years, and more than half suffer from the condition [5,6]. There are two types of dysmenorrhea. Primary dysmenorrhea (PD) presents with lower abdominal pain without evident organic pelvic disease and is more common in younger women after their menstrual cycle is established. Secondary dysmenorrhea (SD) is associated with disorders of the reproductive organs [5,7]. The prevalence of PD in women may be up to six times higher than those with SD [8]. These disorders also vary in Asian women. It has been estimated that 15.8–78.5% of Japanese women suffer from PD [9,10], with more than 60% having moderate to severe lower abdominal cramps [10] and almost half resorting to self-medication [9]. Prevalence rates in Taiwan and Korea were 70.7% and 75.1%, respectively [11,12,13], higher than those reported for women in China, ranging from 41.7 to 56.4% [2,14]. A cross-sectional study in secondary schools in Kuala Lumpur, Malaysia, reported that 79.7% of Malaysian girls, 69.8% of Chinese girls, and 82.4% of Indian girls had suffered from dysmenorrhea [15]. Dysmenorrhea symptoms might differ between White and Asian women. White women generally experience more intense and longer-lasting pain [16].

Stroke is the third leading cause of death globally and one of main causes of disability [17]. Approximately 15 million new stroke patients are diagnosed annually worldwide [18]. With one stroke event every 40 s, it is also the third leading cause of death among women in the United States [19,20]. The Framingham Heart Study found that strokes occur less frequently in women than men in the younger population [21]. Female stroke patients tend to present worse sequelae than male stroke patients [22,23]. A case–control study examining Taiwanese women with dysmenorrhea aged 15–49 years found an increased risk of stroke with age, significant for those aged 30 years and older [24]. Hypertension is also a significant risk factor for stroke, with an adjusted odds ratio of 4.53. However, types of dysmenorrhea were not addressed in these studies.

The complicated pathophysiology of PD is associated with the overexpression of prostaglandin. Prostaglandin is thought to be one of the mediators of chronic vascular inflammation, which has been linked to heart disease and stroke [3,25]. A recent study found that women with PD have an increased risk of ischemic heart disease [26]. Women with PD may also be at higher risk for stroke. To our knowledge, no study has investigated stroke risk specifically for women with PD. Because PD is more prevalent in women than SD, we conducted a study to examine the risk of stroke in women with PD using insurance claims data from Taiwan.

## 2. Methods

### 2.1. Data Source

In this retrospective cohort study, we used the Longitudinal Health Insurance Database (LHID) with claims data of one million insured persons randomly selected from the National Health Insurance Research Database (NHIRD) established by the National Health Insurance Administration of Taiwan. The insurance system was established in 1995 as a mandatory enrollment program, with over 99% of Taiwan’s 23.72 million residents covered. The database contains medical records of outpatients and inpatients and demographic data from 1996 to 2013. Diseases are coded using the International Classification of Diseases, Ninth Revision, Clinical Modification (ICD 9-CM), and the Anatomical Therapeutic Chemical (ATC) classification system. In addition, all identification numbers of insured persons in the claims data were re-coded before being made available to users to protect privacy. This study was approved by the Research Ethics Committee of China Medical University and Hospital in Taiwan (CMUH104-REC2-115 (CR-4)).

### 2.2. Study Population

From LHID claims data, we identified 35,977 women with dysmenorrhea (ICD-9-CM 625.3) newly diagnosed between 2000 and 2010 with at least two consecutive diagnoses as the potential study population. The date of the first dysmenorrhea diagnosis was defined as the index date. Patients with only one diagnosis were not selected to avoid coding and/or medical billing errors. To create the study cohorts, women aged <15 or >40 years with a history of stroke, endometriosis, uterine myoma or pelvic inflammatory disease, hysterectomy, ovariectomy, or cancer, or aspirin use were excluded (Figure 1). We also excluded women with follow-up duration <1 year due to death, stroke, or withdrawal from the insurance. Women aged 41–49 years old were also excluded to avoid the potential impact of premenopausal and early menopause. Excluding women with diagnoses of obvious gynecologic conditions resulted in the exclusion of women with SD [27,28]. The same exclusion criteria were applied to women without dysmenorrhea for comparisons. From the remaining 18,812 women with dysmenorrhea and 101,154 women without dysmenorrhea, we established a PD cohort and a comparison cohort matched by age, index date, and propensity score. We randomly assigned an index date for each comparison woman. We estimated the propensity score for each woman using logistic regression to estimate the probability of disease status on the basis of the baseline variables of age; index date; and comorbidities including diabetes (ICD-9 code: 250; A code: A181), hypertension (ICD-9 code: 401–405; A codes: A260 and A269), hyperlipidemia (ICD-9 codes: 272.0, 272.1, 272.2, 272.3, and 272.4), obesity (ICD-9 codes: 278, A183), alcoholism (ICD-9 codes: 291, 303, 305.00, 305.01, 305.02, 305.03, 790.3, and V11.3), arrhythmia (ICD-9 code: 427), thyroid disease (ICD-9 code: 240–246), migraine (ICD-9 code: 346), immune disorders (ICD-9 code: 279), systemic lupus erythematosus (ICD-9 code: 710.0), and rheumatoid arthritis (ICD-9 code: 714.0). All comorbidities were defined before the index date, with at least two consecutive diagnoses.

### 2.3. Outcome

Strokes that occurred shortly after inclusion in the study cohort may not have been associated with the risk factor. To adjust for the effect of immortal time bias, we began the follow-up one year after each individual’s entry into the cohort. The follow-up person-years were counted up to the date the stroke (ICD-9-CM 430–438) was diagnosed, including hemorrhagic stroke (ICD-9-CM 430-432) and ischemic stroke (ICD-9-CM 433–438), or up to the date of withdrawal from the insurance program, or the end of 2013, whichever occurred first. The maximum follow-up period was 13 years.

### 2.4. Statistical Analysis

This study used SAS version 9.4 software (SAS Institute, Cary, NC, USA) to manage the data and perform the statistical analysis. A two-tailed *p*-value of less than 0.05 was considered to be statistically significant. Data analysis first compared the frequency distributions of age, comorbidities, and the use of non-steroidal anti-inflammatory drugs (NSAIDs) between the 2 cohorts. Women who had been prescribed NSAIDs for 10 days or longer were considered users. A chi-squared test was used to test the distribution of categorical variables between the two cohorts. Mean ages with standard deviations were compared between the two cohorts and tested using the *t*-test. We estimated and plotted the cumulative incidence proportions for overall stroke, ischemic stroke, and hemorrhagic stroke using Kaplan–Meier analysis. The log-rank test was used to examine the difference between the two cohorts. The incidence rate of stroke was calculated by dividing the number of stroke cases by the sum of follow-up person-years for each cohort. Cox proportional hazards regression analysis was used to estimate the PD cohort to the comparison cohort hazard ratio (HR) of stroke and the associated 95% confidence interval (CI). Age and comorbidity-associated HRs of stroke were assessed. We presented the Cox model estimated adjusted HR (aHR), which was estimated after controlling for matched pairs. We also presented results separately for ischemic stroke, hemorrhagic stroke, and the two stroke types combined together as the overall stroke. The likelihood ratio test was used to examine the interaction effects between the PD status and age, comorbidities, and NSAID.

## 3. Results

With similar sample sizes in the matched cohorts with and without dysmenorrhea (*n* = 18,783), distributions of age and comorbidities of both cohorts were similar (Table 1). With an average age of 25.5 years, 54% of the study population was aged 15–24 years. Thyroid disorders were the most prevalent among baseline comorbidities in both cohorts, whereas systemic lupus erythematosus and rheumatoid arthritis were the least common. Few women were taking NSAIDs for 10 days or longer.

### 3.1. Overall Stroke

The Kaplan–Meier method estimated cumulative incidence of stroke after a maximum of the 13-year follow-up period was approximately 0.16% higher in the dysmenorrhea cohort than in the comparison cohort (0.88% vs. 0.72%) (log-rank test *p* = 0.010, Figure 2a), mainly contributed to by ischemic stroke (Figure 2b). The incidence rate of stroke was 1.51 times higher in women with dysmenorrhea than in the comparison cohort (6.05 vs. 4.01 per 10,000 person-years or 99 vs. 65 cases), with an aHR of 1.51 (95% CI 1.11–2.06) after adjustment for matched pairs (Table 2). The difference in incidence rates between cohorts was greater in the 25–40-year-old group (9.71 − 6.13 = 3.58 per 10,000 person-years) than in the 15–24-year-old group (3.02 − 2.25 = 0.77 per 10,000 person-years). The Cox method estimated PD cohort to comparison cohort HRs showed that none of the comorbidities had a significant role associated with stroke. There were no significant interaction effects between age and PD status and between each comorbidity status and PD status.

### 3.2. Ischemic Stroke

The Kaplan–Meier analysis showed that the cumulative incidence of ischemic stroke during the follow-up period was approximately 0.15% higher in the dysmenorrhea cohort than in the comparison cohort (0.65% vs. 0.48%) (log-rank test *p* = 0.012, Figure 2b). Table 3 shows that the incidence rate of ischemic stroke in women with dysmenorrhea was 1.62 times higher than that in the comparison cohort (4.40 vs. 2.71 per 10,000 person-years or 72 vs. 44 cases), with an aHR of 1.61 (95% CI, 1.11–2.33) after adjustment for matching pairs. In addition, the difference in incidence rates between the two cohorts was greater in the 25–40-year-old group (7.82 − 4.08 = 3.74 per 10,000 person-years) than in the 15–24-year-old group (1.56 − 1.58 = −0.02 per 10,000 person-years). The ischemic stroke incidence rates in women with comorbidities were not all higher in women with PD. The Cox method estimated HRs also demonstrated that none of the comorbidities were significantly associated with stroke. There were no significant interaction effects between comorbidities and PD status.

### 3.3. Hemorrhagic Stroke

The Kaplan–Meier analysis shows that the cumulative incidence of hemorrhagic stroke during the follow-up was slightly higher in the dysmenorrhea cohort than in the comparison cohort by the follow-up year of 13 (log-rank test *p* = 0.371, Figure 2C).

Table 4 also shows that the hemorrhagic stroke incidence rate was slightly higher in women with dysmenorrhea than in the comparison cohort (1.65 vs. 1.29 per 10,000 person-years), with an aHR of 1.30 (95% CI, 0.74–2.29). The hemorrhagic stroke was not associated with the comorbidities.

## 4. Discussion

In our study, the distributions of age and comorbidities at baseline were similar in the age- and propensity-score-matched PD cohort and the comparison cohort. We found that more than half of the PD patients were younger women aged 15–24 years, consistent with previous studies [6]. The incidence of stroke increased with age in our study population, which is consistent with the stroke in young women in the Netherlands [29]. We found age had an important role in the development of stroke. The difference of stroke incidence rates between the older and younger groups was near twofold greater in in the PD cohort than in the comparison group (6.69 vs. 3.88 per 10,000 person-years). The aHR of 1.34 for stroke was insignificant in women aged 15–24 years with PD. For women aged 25–40 years, the aHR of 1.58 for stroke was significant in women with PD, indicating that the older women with PD were at a higher stroke risk.

Some comorbidities increased stroke incidence and might be higher in the PD cohort than in the comparison cohort. However, the Cox method estimated PD cohort to comparison cohort HRs showed that none of these comorbidities were significant factors associated with stroke. The propensity score matching reduced the potential bias in stroke development associated with comorbidities.

In our study population, most prevalence rates of baseline comorbidities were less than 5.00%. Among the baseline comorbidities, the prevalence rates of thyroid disease were the highest in both cohorts, slightly lower in the PD cohort than in the comparisons (6.07% versus 6.19%). The disease was associated with a 2.3-fold higher incidence rate of stroke in the PD cohort than in the comparison cohort (7.14 vs. 3.04 per 10,000 person-years), but the numbers of associated stroke cases were few (seven versus three) and the aHR of depression for the PD cohort was not significant. Our data showed that the prevalence of hypertension in women with PD was slightly higher than that in the comparison cohort, which is consistent with the study conducted on women in Tianjin, China [30]. Women with hypertension had a higher incidence of stroke than women without hypertension in both cohorts. Interestingly, the hypertensive women in the PD cohort had a lower incidence of stroke than the comparison group. We suspect that the hypertensive women in the PD cohort could receive more medical attention to better control hypertension [31,32]. The benefit is insignificant, with an aHR of 0.51 (95% CI, 0.16–1.61) for stroke among hypertensive women in the PD cohort.

It is well known that ischemic stroke accounts for most strokes, nearly 70% to 80% of all strokes [29,33]. In the present study, ischemic stroke was also the main type of stroke in both cohorts. The PD cohort had a higher proportion of ischemic stroke than the comparison cohort (72.7% vs. 67.7%, or 72/99 vs. 44/66). The aHR of ischemic stroke was 1.61 (95% CI 1.11–2.33) for the PD cohort, mainly due to an elevated incidence rate in the older women. Women with PD may have increased thrombotic or embolic events that reduce blood flow to the brain [34,35]. Our data also showed that women with PD may be at higher risk for hemorrhagic stroke, but this was not significant, probably because of the small number of cases. It is unclear as to whether hypertension is associated with hemorrhagic events in women with PD. We suspect that the increased risk of stroke in women with PD is related to an imbalance of prostaglandins [36,37].

Women suffering from dysmenorrhea pain may experience a decreased quality of life [38,39]. NSAIDs may be prescribed to relieve the pain [21,28]. Studies have reported that long-term users of NSAIDs are at an increased risk of stroke, particularly hemorrhagic stroke [23,28,38,40,41]. A recent study examining the risk of stroke associated with NSAID uses for dysmenorrhea found that patients who took the medicine for 13 or more days per month were at an elevated risk of stroke. Those taking NSAIDs less than 13 days per month were at a lower risk of stroke with an aHR of 0.51 (95% CI 0.13–2.10) [42]. However, very few women had been prescribed NSAIDs for more than 10 days during the period with PD in this study. Therefore, the impact of this drug was not considered in our study.

There are some limitations in our study. First, PD is a common complaint among young women in our study. The finding is consistent with other Asian populations [9,15]. However, no study has investigated the stroke risk associated with PD for other Asian women. Second, it has been noted that the imbalance of hormones is not only related to dysmenorrhea but also affects the mechanism of estrogen in neuroprotection, reducing the risk of stroke [43]. Thus, women with dysmenorrhea have a higher risk of stroke than women in general. However, the insurance claims data provided no information on the laboratory data of hormones for evaluating the impact of estrogen. Third, a previous focus group study in Taiwan reported that young PD female patients used various self-care strategies, including diet, herbal remedies, and other complementary therapies [44]. Unfortunately, we could not evaluate the impact of these self-care treatments because they are not available in our database as well. Four, certain lifestyle information (smoking, alcohol consumption, exercise, and body mass index) is unavailable in the claims data and, therefore, could not be further adjusted in this study [45,46]. Finally, information on the severity of dysmenorrhea is not available to assess whether women with severe pain are at higher risk for stroke.

## 5. Conclusions

In this propensity score-matched follow-up study, we controlled for the effects of other comorbidities that might be associated with the risk of stroke. Our data showed that, with the exception of age, none of the comorbidities had a significant association with stroke risk for women with PD. Women with PD had an aHR of 1.51 for stroke compared to women without PD, mainly because of ischemic stroke. The age-specific data showed that the aHR of developing stroke in women with PD was significant for women aged between 25 and 40 years old, but not for those aged from 15 to 24, suggesting that healthcare providers may need to counsel women with dysmenorrhea with care strategies, particularly for older individuals.

## Figures and Tables

**Figure 1 jpm-13-00114-f001:**
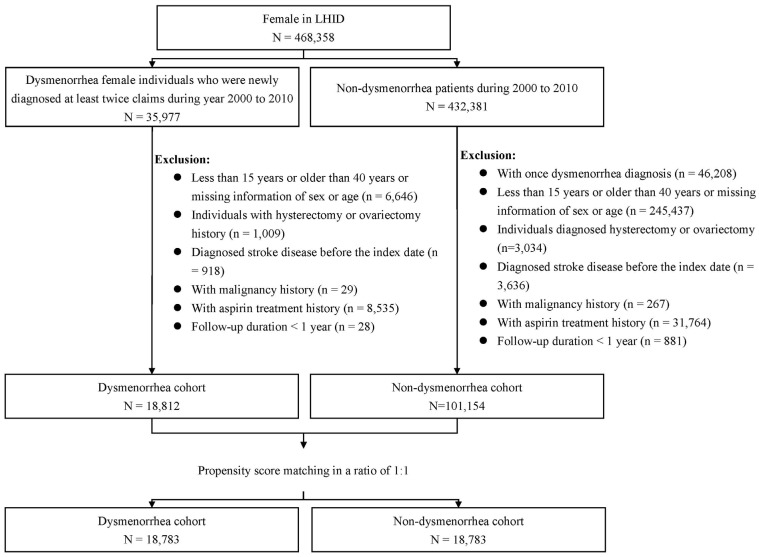
Flow chart for establishing study cohorts.

**Figure 2 jpm-13-00114-f002:**
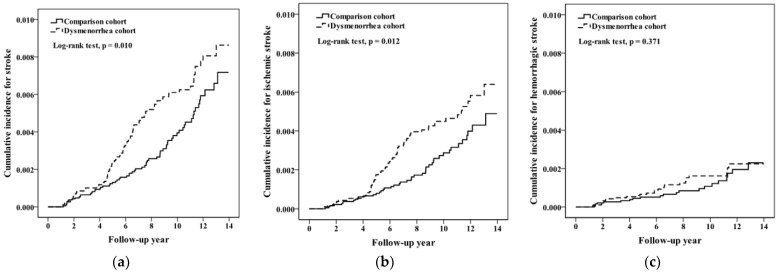
Kaplan–Meier method estimated cumulative incidence for (**a**) stroke, (**b**) ischemic stroke, and (**c**) hemorrhagic stroke in dysmenorrhea and comparison cohorts.

**Table 1 jpm-13-00114-t001:** Distributions of age, comorbidities, and NSAID use compared between cohorts with and without primary dysmenorrhea.

Variable	Primary Dysmenorrhea				*p*-Value *
	No (N = 18,783)		Yes (N = 18,783)		
	*n*	%	*n*	%	
Age group (years)					0.99
15–19	524	26.8	5010	26.7	
20–24	5104	27.2	5134	27.3	
25–29	3635	19.4	3618	19.3	
30–40	5020	26.7	5021	26.7	
Mean (SD) ^a^	25.5	(6.97)	25.5	(6.96)	0.93 ^a^
Baseline comorbidities					
Diabetes mellitus	176	0.94	196	1.04	0.30
Hypertension	176	0.94	195	1.04	0.32
Hyperlipidemia	252	1.34	265	1.41	0.56
Obesity	90	0.48	116	0.62	0.07
Alcoholism	40	0.21	47	0.25	0.45
Arrhythmia	483	2.57	473	2.52	0.74
Thyroid diseases	1162	6.19	1141	6.07	0.65
Migraine	640	3.41	640	3.41	1.00
Immune disorders	39	0.21	50	0.27	0.24
Systemic lupus erythematosus	12	0.06	22	0.12	0.09
Rheumatoid arthritis ^b^	2	0.01	6	0.03	0.29 ^b^
NSAID use					0.31
No	18,724	99.7	18,716	99.6	
Yes	59	0.3	67	0.4	

* chi-squared test. ^a^
*t*-test. ^b^ Fisher’s exact test. NSAID, non-steroidal anti-inflammatory drug.

**Table 2 jpm-13-00114-t002:** Overall number of stroke events and incidence rates in primary dysmenorrhea and comparison cohorts by age, comorbidity, and NSAID use, and Cox model estimated primary dysmenorrhea cohort to comparison adjusted hazard ratio and 95% confidence interval.

Variables	Primary Dysmenorrhea						Hazard Ratio		*p* for Interaction
	No			Yes					
	(N = 18,783)			(N = 18,783)					
	Event, *n*	Person Years	Incidence Rate	Event, *n*	Person Years	Incidence Rate	Crude (95% CI)	Adjusted (95% CI)	
Total	65	162,247	4.01	99	163,625	6.05	1.52 (1.07–2.09) **	1.51 (1.11–2.06) **	
Age, year									0.63
15–24	20	88,807	2.25	27	89,478	3.02	1.36(0.73–2.50)	1.34 (0.75–2.40)	
25–40	45	73,440	6.13	72	74,147	9.71	1.61(1.08–2.32) *	1.58 (1.09–2.28) *	
Diabetes mellitus									0.96
No	62	160,771	3.86	99	161,922	6.11	1.60 (1.10–2.19) **	1.58 (1.16–2.17) **	
Yes	3	1476	20.33	0	1704	0.00	NA	NA	
Hypertension									0.06
No	57	160,814	3.54	94	161,963	5.80	1.65 (1.16–2.30) **	1.64 (1.18–2.27) **	
Yes	8	1433	55.84	5	1663	30.07	0.54 (0.13–1.66)	0.51 (0.16–1.61)	
Hyperlipidemia									0.34
No	64	160,444	3.99	94	161,565	5.82	1.45 (1.06–2.00) *	1.46 (1.06–2.00) *	
Yes	1	1803	5.55	5	2060	24.27	4.29 (0.29–44.2)	4.03 (0.47–34.7)	
Obesity									0.74
No	64	161,542	3.96	96	162,682	5.90	1.50 (1.02–2.11) *	1.49 (1.09–2.03) *	
Yes	1	705	14.18	3	943	31.81	2.31 (0.18–23.4)	2.01 (0.20–20.0)	
Alcoholism									0.97
No	63	161,981	3.89	99	163,267	6.06	1.58 (1.10–2.23) **	1.56 (1.14–2.13) **	
Yes	2	266	75.18	0	358	0.00	NA	NA	
Arrhythmia									0.96
No	62	158,292	3.92	99	159,573	6.20	1.60 (1.13–2.22) **	1.58 (1.16–2.17) **	
Yes	3	3955	7.59	0	4052	0.00	NA	NA	
Thyroid disease									0.51
No	62	152,380	4.07	92	153,816	5.98	1.48 (1.05–2.11) *	1.46 (1.06–2.01) *	
Yes	3	9867	3.04	7	9809	7.14	2.39 (0.44–9.86)	2.41 (0.64–9.04)	
Migraine									0.94
No	62	157,425	3.94	94	158,629	5.93	1.53 (1.01–2.16) *	1.50 (1.09–2.07) *	
Yes	3	4822	6.22	5	4996	10.01	1.64 (0.26–7.11)	1.58 (0.37–6.71)	
Immune disease									1.00
No	65	161,878	4.02	99	163,037	6.07	1.53 (1.08–2.10) **	1.51 (1.11–2.06) **	
Yes	0	370	0.00	0	588	0.00	NA	NA	
NSAID use									1.00
No	65	161,735	4.02	99	163,041	6.07	1.53 (1.07–2.11) **	1.51 (1.11–2.06) **	
Yes	0	512	0.00	0	584	0.00	NA	NA	

Abbreviation: incidence rate, per 10,000 person-years; CI, confidence interval; NSAID, non-steroidal anti-inflammatory drug; NA, not applicable. Immune disease included immune disorders, systemic lupus erythematosus, and rheumatoid arthritis. *p*-values for hazard ratio: * < 0.05; ** < 0.01; *p* for interaction: *p*-value for interaction between dysmenorrhea status and stratified covariate.

**Table 3 jpm-13-00114-t003:** Number of ischemic stroke events and incidence rates in in primary dysmenorrhea and comparison cohorts by age, comorbidity, and NSAID use, and Cox model estimated primary dysmenorrhea cohort to comparison adjusted hazard ratios and 95% confidence intervals.

Variables	Primary Dysmenorrhea						Hazard Ratio		*p* for Interaction
	No			Yes					
	(N = 18,783)			(N = 18,783)					
	Event, *n*	Person Years	Incidence Rate	Event, *n*	Person Years	Incidence Rate	Crude (95%CI)	Adjusted (95%CI)	
Total	44	162,247	2.71	72	163,625	4.40	1.63 (1.07–2.35) *	1.61 (1.11–2.33) *	
Age, year									0.14
15–24	14	88,807	1.58	14	89,478	1.56	0.98 (0.44–211)	0.99 (0.47–2.07)	
25–40	30	73,440	4.08	58	74,147	7.82	1.91 (1.18–2.97) **	1.89 (1.22–2.92) **	
Diabetes mellitus									0.97
No	41	160,771	2.55	72	161,922	4.45	1.75 (1.12–2.60) **	1.73 (1.19–2.52) **	
Yes	3	1476	20.33	0	1704	0.00	NA	NA	
Hypertension									0.16
No	38	160,814	2.36	67	161,963	4.14	1.76 (1.12–2.63) **	1.73 (1.17–2.57) **	
Yes	6	1433	41.88	5	1663	30.07	0.72 (0.15–2.40)	0.68 (0.20–2.31)	
Hyperlipidemia									0.98
No	44	160,444	2.74	67	161,565	4.15	1.51 (1.01–2.32) *	1.50 (1.03–2.18) *	
Yes	0	1803	0.00	5	2060	24.27	NA	NA	
Obesity									0.78
No	43	161,542	2.66	69	162,682	4.24	1.58 (1.08–2.30) *	1.58 (1.08–2.30) *	
Yes	1	705	14.18	3	943	31.81	2.24 (0.18–22.3)	2.01 (0.20–20.0)	
Alcoholism									0.98
No	42	161,981	2.59	72	163,267	4.41	1.71 (1.04–2.51) **	1.68 (1.16–2.45) **	
Yes	2	266	75.18	0	358	0.00	NA	NA	
Arrhythmia									0.97
No	41	158,292	2.59	72	159,573	4.51	1.75 (1.10–2.61) **	1.73 (1.18–2.52) **	
Yes	3	3955	7.59	0	4052	0.00	NA	NA	
Thyroid disease									0.93
No	42	152,380	2.76	69	153,816	4.49	1.63 (1.07–2.40) *	1.61 (1.10–2.35) *	
Yes	2	9867	2.03	3	9809	3.06	1.52 (0.20–9.01)	1.50 (0.25–8.93)	
Migraine									0.63
No	42	157,425	2.67	67	158,629	4.22	1.61 (1.02–2.37) *	1.57 (1.07–2.30) *	
Yes	2	4822	4.15	5	4996	10.01	2.42 (0.40–13.2)	2.39 (0.46–12.4)	
Immune disease									1.00
No	44	161,878	2.72	72	163,037	4.42	1.63 (1.04–2.41) *	1.61 (1.11–2.33) *	
Yes	0	270	0.00	0	588	0.00	NA	NA	
NSAID use									1.00
No	44	161,735	2.72	72	163,041	4.42	1.63 (1.01–2.44) *	1.61 (1.11–2.33) *	
Yes	0	512	0.00	0	584	0.00	NA	NA	

Abbreviation: incidence rate, per 10,000 person-years; CI, confidence interval; NSAID, non-steroidal anti-inflammatory drug; NA, not applicable. Immune disease included immune disorders, systemic lupus erythematosus, and rheumatoid arthritis. *p*-values for hazard ratio: * < 0.05; ** < 0.01; *p* for interaction: *p*-value for interaction between dysmenorrhea status and stratified covariate.

**Table 4 jpm-13-00114-t004:** Number of hemorrhagic stroke events and incidence rates in in primary dysmenorrhea and comparison cohorts by age, comorbidity, and NSAID use, and Cox model estimated primary dysmenorrhea cohort to comparison adjusted hazard ratios and 95% confidence intervals.

Variables	Primary Dysmenorrhea						Hazard Ratio		*p* for Interaction
	No			Yes					
	(*n* = 18,783)			(*n* = 18,783)					
	Event	Person Years	Incidence Rate	Event	Person Years	Incidence Rate	Crude (95%CI)	Adjusted (95%CI)	
Total	21	162,247	1.29	27	163,625	1.65	1.32 (0.70–2.37)	1.30 (0.74–2.29)	
Age, year									0.17
15–24	6	88,807	0.68	13	89,478	1.45	2.22 (0.70–5.98)	2.18 (0.83–5.76)	
25–40	15	73,440	2.04	14	74,147	1.89	0.96 (0.41–2.33)	094 (0.46–1.93)	
Diabetes mellitus									1.00
No	21	160,771	1.31	27	161,922	1.67	1.33 (0.68–2.40)	1.30 (0.74–2.29)	
Yes	0	1476	0.00	0	1704	0.00	NA	NA	
Hypertension									0.98
No	19	160,814	1.18	27	161,963	1.67	1.36 (0.78–2.69)	1.34 (0.81–2.58)	
Yes	2	1433	13.96	0	1663	0.00	NA	NA	
Hyperlipidemia									0.98
No	20	160,444	1.25	27	161,565	1.67	1.40 (0.67–2.55)	1.37 (0.77–2.43)	
Yes	1	1803	5.55	0	2060	0.00	NA	NA	
Obesity									1.00
No	21	161,542	1.30	27	162,682	1.66	1.35 (0.66–2.70)	1.30 (0.74–2.29)	
Yes	0	705	0.00	0	943	0.00	NA	NA	
Alcoholism									1.00
No	21	161,981	1.30	27	163,267	1.65	1.34 (0.70–2.69)	1.30 (0.74–2.29)	
Yes	0	266	0.00	0	358	0.00	NA	NA	
Arrhythmia									1.00
No	21	158,292	1.33	27	159,573	1.69	1.32 (0.71–2.57)	1.30 (0.74–2.29)	
Yes	0	3955	0.00	0	4052	0.00	NA	NA	
Thyroid disease									0.28
No	20	152,380	1.31	23	153,816	1.50	1.19 (0.52–2.79)	1.15 (0.63–2.09)	
Yes	1	9867	1.01	4	9809	4.08	4.15 (0.34–44.9)	4.27 (0.54–34.1)	
Migraine									0.98
No	20	157,425	1.27	27	158,629	1.70	1.34 (0.66–2.72)	1.36 (0.77–2.42)	
Yes	1	4822	2.07	0	4996	0.00	NA	NA	
Immune disease									1.00
No	21	161,878	1.30	27	163,037	1.66	1.32 (0.64–2.57)	1.30 (0.74–2.29)	
Yes	0	370	0.00	0	588	0.00	NA	NA	
NSAID use									1.00
No	21	161,735	1.30	27	163,041	1.66	1.29 (0.66–2.59)	1.30 (0.74–2.29)	
Yes	0	512	0.00	0	584	0.00	NA	NA	

Abbreviation: incidence rate, per 10,000 person-years; CI, confidence interval; NSAID, non-steroidal anti-inflammatory drug; NA, not applicable. Immune disease included immune disorders, systemic lupus erythematosus, and rheumatoid arthritis. *p* for interaction: *p*-value for interaction between dysmenorrhea status and stratified covariate.

## Data Availability

All data are incorporated into the article.

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
