# Peer review of "Stroke Risk in Young Women with Primary Dysmenorrhea: A Propensity-Score-Matched Retrospective Cohort Study"

_jpm, 2023, doi:10.3390/jpm13010114_

Round 1

Reviewer 1 Report

In the manuscript, Dr. Yeh and coauthors conducted a propensity score matched retrospective study to examine the risk of stroke in women with PD using insurance claims data from Taiwan. and concluded that women of reproductive age with PD are at increased risk for ischemic stroke. However, the study design and analysis method are questionable and insufficient supporting the conclusion. And the manuscript is not well-written with low quality. I have multiple comments are as follows:

1.       Please clarify the inclusion/exclusion criteria to avoid confusion, for example:

a.     In Figure 1, it states “Follow-up duration < 1 year”, while in section 2.2, the statement is “We further excluded those who died or had a stroke within one year after the diagnosis of dysmenorrhea”. Those criteria are not identical to each other

b.     I would suggest author providing more justification for the criteria “exclude those who died or had a stroke within one year after the diagnosis of dysmenorrhea”. Referring related published reports could be helpful. It is not convincing enough by mentioning the effect of immortal time bias only.

c.     Given that there is no subject with dysmenorrhea diagnosed, please explain how the criteria “exclude those who died or had a stroke within one year after the diagnosis of dysmenorrhea” could be applied for those without dysmenorrhea, if “The same exclusion criteria used for the women with dysmenorrhea were applied to those without dysmenorrhea to make comparisons. “

2.       In section 2.2, as the authors stated, “We estimated the propensity score for each woman using logistic regression to estimate the probability of the treatment assignment with baseline variables of ...”. What treatment assigned for target cohort and why the probability of the treatment assignment should be estimated?

3.       Please clarify the detailed method/models used to examine the interaction effect of age, comorbidities, and NSAID use with dysmenorrhea, e.g. whether covariates adjusted in the Cox model, whether the interaction term between dysmenorrhea vs. covariates was tested or covariates were tested as stratification factors?

4.       Please distinguish the definition of “incidence”, “incidence rate”, “risk”, and/or “hazard” and be consistent when describing the results.

5.       In section 3.1, the authors mentioned “Patients with dysmenorrhea had a significantly higher risk of stroke than those without comorbidity or NSAID use.” By using the wording “significantly higher”, please clarify the analysis method and corresponding results for this statement.

6.       Please consider rephrasing the caption of Table 2 and reorganizing it. It’s difficult for audience to read. It would be suggested to include IR difference with CI and p value as well as p value for aHR in the table. And modify Table 3 and 4 accordingly.

7.       In section 3.3, “Table 4 shows the hemorrhagic stroke incidence was slightly higher in dysmenorrhea women than in the comparison cohort (1.65 vs. 1.29 per 10,000 person-years), with an HR of 1.30 (95% CI, 0.74-2.29).” Is it an “HR” or “aHR”? Please clarify the definition and be consistent when describing the results. In section 2.2 “HR” was defined without any results or discussion presented in Results section.

8.       In Discussion, please clarify the analysis method and corresponding results for the statement in line 11-13

9.       In Discussion, the statement in line 13-15 is not clear to me. A comprehensive discussion would be necessary.

10.    The sentence in line 16-18 is not clear to me. And cannot identify the results “2.3-fold higher incidence but not significant”.

11.    Please provide reference for the sentence in line 23-24

12.    please clarify the analysis method and corresponding results for the sentence in line 26-27

13.    In Discussion line 32-33, aHR is hazard ratio but not risk difference.

14.    In Discussion line 52-56, please be cautious with any conclusion without well-planned analysis & corresponding results support

15.    In Discussion line 56-57, the authors provide insufficient evidence for the conclusion. In addition, this sentence is not related to previous statements.

16.    In Conclusion line 72-73, please interpret aHR correctly, which is hazard ratio but not risk difference.

17.    There is no sufficient evidence to support the conclusion in line 73-74

18.    The sentence in line 74-75 is not relevant to main conclusion of the manuscript.

There are quite a bit grammatical errors and ambiguity throughout the manuscript, please try going through the manuscript and revise as much as possible. 

Author Response

Thank you for your so precious review. We uploaded the response in the additional paper. Please check. Thank you very much.

Best regards!

Chunghsin Yeh

Reviewer 2 Report

This article is an interesting topic focused on significance of content that taking into account specifically pathologies of women . These results are important to understand the women´s psysiology and highlighting  studies with a gender perspective.    

However, the authors should review results the tittle of figures 2A and 2b in the results section to properly reference them. In addition, in the discussion section the authors mentioned the importance of  imbalance of hormones to explain the dysmenorrhea but they did not show any data about this in the sample. The authors also underlined that estrogen may play an important neuroprotection role in women. Can they explained why did not consider this clinical data about hormonal information? or can be mentioned this limitation in this study?

Author Response

(The authors gave the same response as above.)

Round 2

Reviewer 1 Report

The authors have addressed majority of my concerns. In addition, a revision is recommended to improve the quality of the article addressing the concerns and questions listed below:

1.     It’s clear now about the inclusion/exclusion criteria that patients with “Follow-up duration < 1 year” were excluded. Could the authors clarify the definition of baseline and/or Day 1 for survival analysis? Based on the statement in section 2.1 “The first date of dysmenorrhea diagnosis was defined as the index date.”, is this index date considered as Day 1 for downstream analysis? What time is defined as Day 1 for subjects without dysmenorrhea? And justification for Day 1 definition, which makes a fair comparison in the downstream analysis, is recommended.

2.     It’s still not clear for me how the interaction effects between the PD status and age, comorbidities, and NSAID use were tested. But it is ok, given that multiple methods could be used for interaction test.

3.     The captions of Table 2-4 are still confused. Please consider revising them appropriately, for example “Incidence rates and aHR estimates for stroke by age and comorbidity subgroups in primary dysmenorrhea vs. comparison cohorts”. The authors have ignored my comments on table 2-4 regarding the incidence rate and aHR. The incidence rate ratio was calculated instead of incidence rate difference, which also makes sense. However, the table might be difficulty to read for audience. It would be suggested to include IR ratio with CI and p value as well as p value for aHR in the table to avoid any confusion. Column “Event” and “Person Years” could be combined if necessary. If there are too many details in one table, then separate table can be provided. Or p values can be omitted but presenting * (**) with the significant level <0.05 (<0.0q), which is also fine.

4.     The authors defined HR and stated “Cox proportional hazards regression analysis was used to estimate the PD cohort to the comparison cohort hazard ratio (HR) of stroke and the associated 95% confidence interval (CI).” in section 2.4, but not present crude HR with CI, which seems redundant but acceptable.

5.     In discussion, the revised sentence “However, the Cox method estimated PD cohort to comparison cohort aHRs showed that none of these comorbidities were significant factors contributing to the development of stroke in women with PD.” is not clear to me. To claim the conclusion “none of these comorbidities were significant factors contributing to the development of stroke in women with PD”, the effects of comorbidities on stroke should be tested in women with PD. However, there is no such results reported. On the other hand, aHRs were estimated and tested by comorbidity subgroups to assess the effects of PD status (with/without PD) on stroke, which is not sufficient to support the conclusion. Similar statements were found in section 3.1 “In the stratified analysis associated with comorbidities, the Cox method estimated PD cohort to comparison cohort aHR showed that none of the comorbidities had a significant role associated with stroke for women with PD.” and line 30-33 “The Cox method estimated PD cohort to comparison cohort aHRs for women with comorbidities showed that none of the comorbidities were significantly associated with the development of stroke for the PD cohort, revealing that PD may be an independent risk factor.” as well.

     In addition, for several comorbidity subgroups, e.g. Hypertension, Hyperlipidemia, Obesity, Thyroid disease in Table 2..., aHRs show with significant/insignificant p values among subjects with/without certain comorbidity. Further discussion on those results is welcomed.

6.     In section 3.2 “The comorbidity-specific analysis also revealed no significant association between any comorbidity and stroke.”, could the authors guide me where the results located for this? The p values for interaction effects of comorbidity and PD status on stroke were reported. But I didn’t see the results for effects of comorbidity on stroke.

Author Response

Respected Reviewer:

   Thank you for the 2nd Review Report Form 2 and granting us the 2nd opportunity to revise the above referred article.

     We have responded to the comments point by point raised by the Reviewer. We were particularly concerning the precise comments, which mentioned the potential association between dysmenorrhea and stroke risk. We clarified the index day in both groups (dysmenorrheal and comparison). Additionally, we revised and rephrased those tables and sentences according to the Reviewer mentioned. Crude HRs are also added to Tables 2-4.

   We now have completed the revision, responding to comments of Reviewer’s point by point. We are now submitting all materials to you for your review and consideration of our study.

   The manuscript has been proof read by a professor in the USA. All authors have agreed the submission of the revised version. No materials have been submitted to other journal.      

Best Regards! 

Chunghsin Yeh, MD, PhD
